# Stability Evaluation for Heat Tolerance in Lettuce: Implications and Recommendations

**DOI:** 10.3390/plants13111546

**Published:** 2024-06-03

**Authors:** Maryanne C. Pereira, Nara O. S. Souza, Warley M. Nascimento, Giovani O. da Silva, Caroline R. da Silva, Fabio A. Suinaga

**Affiliations:** 1Conselho dos Exportadores de Café (CECAFE), Av. Nove de Julho, 4865, Torre A, Conjunto 61, São Paulo 01407-200, SP, Brazil; mcosta.pereira04@gmail.com; 2Faculdade de Agronomia e Medicina Veterinária (FAV), Campus Darcy Ribeiro, Universidade de Brasília (UnB), ICC-Sul, Asa Norte, Brasília 70910-900, DF, Brazil; narasouza@unb.br (N.O.S.S.); carolinerosaxp@gmail.com (C.R.d.S.); 3Embrapa Hortaliças (CNPH), Rodovia BR 060 Km 9—Samambaia Norte, Brasília 70351-970, DF, Brazil; warley.nascimento@embrapa.br (W.M.N.); giovani.olegario@embrapa.br (G.O.d.S.)

**Keywords:** *Lactuca sativa* L., global warming, abiotic stress

## Abstract

Lettuce is an important cool-temperature crop, and its principal abiotic stress is low heat tolerance. Lettuce production has become more challenging in the context of global warming changes. Hence, the main objective of this research was to investigate the relationship between stability and lettuce heat tolerance. Field and greenhouse trials were run in 2015 (summer) and 2016 (fall and spring). The environments were composed of a combination of season and place (field, glass, and plastic greenhouse), and the assessed genotypes were BRS Leila and Mediterrânea, Elisa, Everglades, Simpson, and Vanda. Statistical analysis showed a significant effect (*p* < 0.05) of environments (E), genotypes (G), and the GEI. BRS Leila, Elisa, and BRS Mediterrânea showed the greatest means to the first anthesis in suitable environments (milder temperatures). Among these cultivars, BRS Mediterrânea was the most stable and adapted to hot environments. The environmental conditions studied in this research, mainly high temperatures, could become a reality in many lettuce-producing areas. Therefore, the results can help indicate and develop lettuce varieties with greater heat tolerance.

## 1. Introduction

Lettuce (*Lactuca sativa* L.) is known worldwide as the most important leafy vegetable. Its consumption is associated with preparing salads that explore the flavors and textures of its leaves, and from a nutritional perspective, it has a high fiber content and low energy value [1]. As stated by Pitrat [2], this species originated from Europe and southwest Asia and is adapted to temperatures ranging from 7 °C to 25 °C [3]. 

In suboptimal temperature conditions, mainly when the temperature is above 30 °C, the leaves become sharpened and have longer internodes, which decreases their commercial value [4]. In addition, high temperatures cause rib discoloration, tip burning, premature bolting, and ribbiness [5,6]. Investigating the effects of heat stress on lettuce, Zhao et al. [7] found changes in the plant biomass and leaf shape. The leaf and stalk length increased significantly, and the leaves became narrower and longer. In addition, after heat stress, the content of chlorophyll and osmotic adjustment substances (proline and soluble sugar) decreased in the early stage and increased in the later stage. Moreover, the activity of antioxidant enzymes (catalase, peroxidase, and superoxide dismutase) and malondialdehyde showed similar patterns.

Besides the biological aspects, it is essential to consider global warming’s effects on growing lettuce [7]. Among all the adverse effects of this phenomenon, extreme heat and drought are considered to be the most harmful to agriculture [8]. In this context, the Intergovernmental Panel on Climate Change (IPCC) estimates that global air temperatures will increase by 0.2 °C per decade, leading to temperatures 1.8–4.0 °C higher than the current level by 2100 [9]. One way to combat the negative effects of high temperatures is to obtain improved lettuce varieties to ensure production in new, challenging environments [10].

Moreover, horticultural crops such as lettuce are often affected by environmental conditions. Thus, some genetic responses are variable and influenced by genotype and environment and their interaction, designated as genotype by environment interaction (GEI). These effects are caused by the joint actions of genotypes grown in different environments, influencing the phenotype [11]. In this context, studying the genetic basis of the GEI is of fundamental relevance for plant breeding in global climate change scenarios [12]. Several statistical procedures have been developed to estimate the effects of the GEI, evaluate adaptability and stability, and facilitate genotype recommendations in multi-environment trials (METs) [13,14,15].

According to Becker and Leon [16], the concept of stability involves several aspects. In this sense, considering the goals of the breeding program and the trait under investigation, stability can be defined as static or dynamic, and the first resembles the biological concept and the latter the agronomic one. In the static or biological concept, a stable genotype has unchanged performance regardless of possible changes in environmental conditions. This genotype generally does not present deviations from the expected characteristic levels. Regarding the dynamic concept, these authors pointed out that a stable genotype has no deviation from predictable responses to environments. Moreover, the dynamic or static concept is related to the nature of the data and test environments, and discriminating them on an absolute basis can be difficult [17].

Nowadays, several stability analysis methods have been designed to evaluate groups of tested genotypic materials. These methods are based on different premises, such as analysis of variance (ANOVA) of the genotype x environment interaction [18] and simple linear regression [19,20]. The disadvantages of these methods are as follows: (1) ANOVA is based on an additive model, which is not accurate in explaining the influence of genotypes and environments on the GEI. As the nature of the GEI is complex, for a complete understanding, it is necessary to use models or protocols more sophisticated than ANOVA [17], and (2) linear regression analysis is not informative if linearity fails. Moreover, it is highly dependent on the group of genotypes and environments included and tends to simplify response models, explaining variations due to interactions in a single dimension [21].

In order to surpass the disadvantages of ANOVA and linear regression analyses, researchers can apply multivariate methods, such as Additive Main effect and Multiplicative Interaction (AMMI) [22] and/or weighted average of absolute scores (WAASB) [23]. Regarding nonparametric methods, Annichiarico [24] and Huenh [25] are examples based on this statistic. An interesting approach was performed using the Lin and Binns method [26] modified by Carneiro [27] through the partitioning of the statistic Pi in favorable (Pi+) and unfavorable (Pi−) environments.

Moreover, in AMMI analysis, an additive model (ANOVA) is adjusted for the general means, the means of G and E, and a multiplicative model (PCA) for the residual of an additive model or the GEI. Hence, AMMI is a good choice for analyzing the GEI in an MET because it estimates the total GEI effect of each genotype and separates the GEI into several interaction effects due to individual environments [22]. Recently, a method called WAASB was proposed, and this methodology combines the features of AMMI and the mixed model BLUP (best linear unbiased predictor) techniques. Besides the advantage of using random effects, WAASB also provides 100% of the variation explained in a bi-dimensional plotted index [23]. Finally, it is crucial to consider that the choice of methodology depends on the experimental data, especially those related to the number of environments available, the precision required, and the type of information desired [21].

Considering the noxious effects of climate change on lettuce production, the main objective of this research was to investigate the relationship between the stability and lettuce heat tolerance.

## 2. Results

In general, the evaluated environments showed three kinds of patterns: (1) high average temperatures (summer), (2) mild temperatures (winter), and (3) intermediate temperatures (fall) (Table 1).

Combined analysis of variance showed highly significant effects (*p* < 0.01) of environments, genotypes, and the genotype by environment interaction (GEI). GEI linear effects and pooled deviations showed highly significant (*p* < 0.01) effects. The AMMI analysis of variance divided the GEI into five principal components (PCs). The first three PCs were significant at *p* < 0.05, and the fourth and fifth PCs were not significant. In this sense, PC1 and PC2 explained 77.90% of the GEI (Table 2).

The importance of the GEI can be illustrated by the substantial difference between the genotypes across the environments, where the heat tolerance, estimated by the number of days to the first anthesis, ranged from 77 (Everglades in environment E3) to 163 (BRS Leila in environment E4). Analogous differences could be noted between E4 (132) and E2 (96) (Table 3). It is also possible to find the environmental index obtained from Eberhardt and Russell in Table 3 [19]. In this sense, E1, E2, E3, and E9 were classified as unfavorable environments, and, on the other hand, the remaining environments were classified as favorable.

It is possible to note that, as shown in Table 4, none of the evaluated cultivars had regression coefficients that were statistically (*p* < 0.05) equal to one. The regression coefficients from BRS Mediterrânea (1.30) and Elisa (1.45) were greater than one, and the remaining cultivars had these values less than one. In addition, the heat tolerance of the cultivars BRS Mediterrânea and Vanda were greater than the general average (115 days), and all the variance values of the regression deviations were significant (*p* < 0.01) and greater than unity. Furthermore, the coefficient of determination was greater than 80%, except for the cultivar Vanda (78.43%). The lowest indexesof ecovalence (*W^2^*) were observed in Simpson (53.36), BRS Mediterrânea (76.54), and Elisa (79.94). Regarding Shukla’s stability variance, the lowest $\sigma_{i}^{2}$ values were found in BRS Mediterrânea (577.60), Simpson (591.27), and Everglades (599.60). Lin and Binns’s methodology [26] can also assess the genotype superiority measure through the statistic Pi. Thus, the cultivars with the lowest Pi’s were BRS Leila (4.76), Elisa (5.62), and BRS Mediterrânea (7.11). These cultivars were also in the same rank regarding favorable (Pi+) and unfavorable (Pi−) environments (Table 4).

The dispersion of the evaluated cultivars based on the mean days to the first anthesis and IPCA1 is shown in Figure 1A. The cultivars BRS Leila (130), Elisa (127), BRS Mediterrânea (123), and Vanda (118) showed several days for the first anthesis greater than the general mean (114). On the other hand, the environments with the lowest number of days for the first anthesis were E1 (105), E2 (96), E3 (103), and E9 (110). Moreover, no IPCA1 scores close to zero were found for any of the assessed cultivars. The position of environments and genotypes could be observed on AMMI2 (Figure 1B). In this sense, none of the evaluated cultivars were close to the origin, and additionally, BRS Mediterrânea and Simpson were plotted close to the E1 and E3 biplot. The same trend was observed for BRS Leila and Elisa, which were plotted close to E4 (Figure 1B).

*WAASB* statistics analysis was performed to cluster the genotypes and environments based on the mean values of the first anthesis and stability. In this context, Figure 2 shows the biplot between the *WASB* index and the number of days to the first anthesis. As postulated by Olivoto et al. [23], all the tested environments and genotypes were divided into four quadrants. The genotype Everglades and environment E3 were placed in the first quadrant, which means that they had a performance that was below the average mean. The second quadrant enclosed BRS Leila, Elisa, and Vanda and environment E4, and they had performances above the average for the number of days to the first anthesis. The genotype and environments within the third quadrant were, respectively, BRS Mediterrânea and environments E5 to E8. The fourth quadrant was composed of genotype Simpson and environments E1, E2, and E9. Finally, as a compilation of all the stability statistics, the lowest values for the Average Sum of Ranks could be observed in BRS Mediterrânea (2.50), Elisa (2.67), and BRS Leila (2.67) (Table 4).

## 3. Discussion

The high temperatures observed mainly in environments E2, E3, E5, and E8 were deleterious to the lettuce in this study, affecting its physiology and diminishing the vegetative phase that anticipated the yield. Moreover, this condition influenced flowering earliness and was related to bitterness and several other disorders in the lettuce. Hence, in all possible temperature scenarios, heat tolerance based on late flowering in lettuce is a keystone of breeding programs [4]. As proposed by Liu et al. [29], the main heat tolerance mechanisms in lettuce comprise heat shock proteins (HSPs) and heat shock factors (HSFs), the most studied regulators and factors in heat response. Hence, those authors found two HSP70 genes that significantly influence lettuce heat-induced bolting tolerance. Moreover, high temperatures can induce LsHSP70 to interact with calmodulin, and these effects were associated with the accumulation of gibberellin. Another effect of heat on lettuce is premature bolting, and LsSOC1 promotes this trait [30,31]. Also, LsFT plays a significant role in bolting regulation [30]. It is also essential to consider the impacts of global climate change on food security, especially in species that are poorly adapted to heat, such as cool-season fruits, vegetables, and grains [32]. In this context, the primary strategies for addressing this problem are mitigating the effects and adapting genotypes to these conditions. Examples of the first strategy are techniques that reduce soil temperature or maintain water in the soil. Regarding adaptation, the main strategy is breeding tolerant varieties to abiotic stresses [33].

According to Oroian et al. [34], the highly significant effect of the linear part of the GEI indicated few similarities among the genotypes in the studied environments. They also point out that selecting well-adapted genotypes is more difficult due to the effects of the GEI on genotype performance. In this research, the effects of the environments and the GEI on heat tolerance could cause a decrease in the heritability of this trait [35]. Moreover, a significant GEI for complex traits, such as heat tolerance estimated by days to anthesis in lettuce, preclude an effective selection difficulty, which makes it challenging to obtain new adapted lettuce cultivars [36].

The significant effect of the environment (*p* < 0.01) indicates that significant environmental variations can provide changes in the performance of genotypes [21]. The same authors also explain that when the linear GEI has significant effects (*p* < 0.01), there are differences among the genotype regression coefficients. Moreover, as the pooled deviation was significant, it means that the linear and non-linear effects of stability influenced the phenotypic expression [37]. These findings support using multivariate and nonparametric approaches to study univariate and multivariate statistics to estimate stability.

The highest and lowest mean days to the first anthesis were noted in environments E4 and E2, respectively, and the difference between them was 36 days, the vegetative cycle of leaf lettuce in most Brazilian regions. This finding could be valuable for growers regarding lettuce production in unfavorable environments. As stated by [38], a high variation in trait(s), for example, stability, and adaptability, could help in developing high-tolerant lettuce to bolting and improve its heat tolerance stability performance.

As examined by Pour-Aboughadareh et al. [17], considering the effect of the GEI on breeding programs increases the efficiency of selection processes and identifies genotypes with high adaptability and stability. Hence, estimates of adaptability and stability are important for selecting superior genotypes. According to Eberhardt and Russell’s [19] methodology, genotypes can be clustered regarding their regression coefficients (less than, equal, or greater than 1), the variance of the regression deviations (equal or not to zero), and the magnitude of the coefficient of determination (R^2^ > 80%). In this context, BRS Leila and Elisa tended to be responsive to favorable environments, as they had $\beta_{1i}$ > 1 and a number of days to the first anthesis greater to the mean. On the other hand, a critical remark should be made regarding BRS Mediterrânea and Vanda in terms of their specific adaptation to unfavorable environments and their performance in terms of the days to the first anthesis. None of the evaluated cultivars had broad adaptability ($\beta_{1i}$ = 1) or predictable responses ($\sigma_{di}^{2}$ = 0) according to the recommendations of Eberhardt and Russell. However, this concept was redesigned by Scapim et al. [39] regarding selecting genotypes with a high expression of traits under unfavorable environments, which is more efficient than searching for genotypes with $\beta_{1i}$ = 1. These authors also pointed out that these genotypes ($\beta_{1i}$ = 1) can perform worse under unfavorable environments than those with $\beta_{1i}$ < 1, for example. Instead of $\sigma_{di}^{2}$ = 0, Schmildt and Cruz [40], associated the high trait performance with values of R^2^ greater than 80% to select highly predictable genotypes.

Regarding the AMMI1 biplot cultivars, Everglades and Simpson had the lowest means of days to the first anthesis. This fact could be explained by their origins, where Simpson is an ancient cultivar of lettuce from the XIX century and Everglades was released in 1993 by several crosses of genotypes adapted to mild temperatures [41]. Considering the AMMI2 model, the cultivars BRS Mediterrânea and Simpson were related to E1 and E3; two pathways could explain these results. As the first cultivar showed a high mean of days to the first anthesis, this association could be explained by its production under high temperatures [42]. An opposite trend is valid for Simpson, which is not adapted to high temperatures, and this cultivar was negatively related to the hottest environments. Regarding the WAASB analysis, as Everglades and E3 were placed in the first quadrant, these cultivars and environments had the most considerable role in the GEI [17]. Moreover, the cultivar Everglades can be classified as unstable. In the second quadrant, the cultivars BRS Leila, Elisa, and Vanda could be classified as unstable even though they had a high performance in terms of the number of days to the first anthesis [23]. Moreover, environment E4 could be used to select genotypes tolerant to early flowering. Olivoto et al. [23] also point out that in the third quadrant, low-performance and wide-adapted genotypes are included due to the lower values of WAASB. In this sense, environments E1 and E2 offered a low discrimination power. Indeed, they showed the highest temperatures, which are entirely unsuitable for growing lettuce. The cultivar Simpson could be classified as stable, but as commented before, it is not interesting because of its poor performance, and the cultivars BRS Mediterrânea and Vanda could be broadly adapted with the above mean performance. Considering the environments in the fourth quadrant, they were productive but with a low discrimination power. This fact is consistent with the average temperature in seasons with low temperatures. It is also important to point out that environments E1 to E3 and E9 were considered unfavorable regarding the protocols of Eberhardt and Russell, AMMI, and WAASB.

As stated by Becker and Leon [16], the static concept of stability is helpful for traits, the levels of which have to be maintained at all costs, e.g., for abiotic stresses, quality traits, or disease resistance. For lettuce, the concept of yield is slightly different from that of grains (for example), as the consumption of lettuce is driven by visual aspects and rarely by weight. Hence, heat causes several market problems, such as thin and long leaves, which implies elongated plants, rib disorders, and a bitter taste in the leaves. Frequently, choosing superior genotypes based on a single statistic could be a problematic approach [38]. According to these authors, one way to overcome this challenge is through the Average Sum of Ranks (ASR), where the lowest values indicate high stability. In this sense, BRS Mediterrânea, Elisa, and BRS Leila were the most stable genotypes found in this research. Suinaga et al. [43,44] evaluated lettuce yield components—weight and number of leaves per plant—and BRS Leila and BRS Mediterrânea showed superior performance to the control variety (Vanda). In the end, based on all the statistics analyses performed in this research, BRS Mediterrânea has adapted to various environments and has a good level of stability.

## 4. Materials and Methods

### 4.1. General Conditions

The trials were run during three seasons in two years: 2015, in summer; 2016, in fall; and 2016, in winter. All cultivations were located at Embrapa Hortaliças, Rodovia BR 060 Km 9, Distrito Federal, Brazil. For each season, the trials were run under the following conditions: field, plastic greenhouse, and glass greenhouse. Each environment was composed by combining the season and conditions (Table 1). In all trials, plants were cultivated in 5 L polyethylene pots containing one plant per pot.

Seedlings were produced at Embrapa using 200-cell foam trays filled with the substrate Carolina^®^ (Carolina Soil do Brasil^®^, Santa Cruz do Sul, Rio Grande do Sul, Brazil). These seedlings were transplanted 25 days after planting when the plants reached three to four leaves. The genotypes were *Lactuca sativa* cvs BRS Leila, BRS Mediterrânea, Elisa, Everglades, Simpson, and Vanda, and their genetic origin is explicated in the Appendix A (Table A1). Pots were filled with a mixture of soil and carbonized rice chaff in a proportion of 5:1 with 40 Kg ha^−1^ of N, 60 Kg ha^−1^ of P_2_O_5_, 30 Kg ha^−1^ of K_2_O, and 70 ton ha^−1^ of cattle compost. Four top dressing fertilizations with 40 Kg ha^−1^ of N were carried out in the first 15 days after transplanting and at 20-day intervals from the first.

### 4.2. Experimental Design and Data Collection

The experimental design involved completely randomized blocks with four replications, and the experimental plot was represented by ten 5 L pots containing the genotypes mentioned above. As stated by Lafta et al. [6], the time lapse to anthesis can indirectly assess heat tolerance in lettuce. Hence, the heat tolerance of the lettuce genotypes was evaluated by counting the number of days until the first anthesis. For each genotype, the average number of days until the first anthesis was calculated.

### 4.3. Statistical Analysis

As indicated in Eberhart and Russell’s analysis, the linear regression model used for this methodology was as follows:$$Y_{ij}=\;\beta_{0i}+\beta_{1i}I_{j}+\delta_{ij}+\bar{\epsilon_{ij}}$$
where $Y_{ij}$ is the observed mean of genotype *i* in environment *j*; $\beta_{0i}$ is the general mean of genotype *i*; $\beta_{1i}$ is the coefficient of regression of genotype *i* and $I_{j}$ is the environmental index *j*; $\delta_{ij}$ is the deviation of the regression of genotype *i* in environment *j*; and $\bar{\epsilon_{ij}}$ is the mean error associated with the average. In addition, $I_{j}$ was calculated as follows:$$I_{j}=\bar{Y_{.i}}-{\bar{Y_{..}}\;with\sum_{j=1}^{n}I}_{j}=0\;$$
where *n* is the number of environments.

Considering Wricke’s ecovalence [18], the statistic *W^2^* was calculated as follows:$$W^{2}={\sum \left(X_{ij\;}-{\bar{X}}_{i.}-\;{\bar{X}}_{.j}+{\bar{X}}_{..}\right)}^{2}$$
where *X_ij_* is the trait of genotype ith in environment jth; *X_i_*_._ is the mean trait of genotype ith; *X_.j_* is the mean trait of environment jth; and *X*_.._ is the trait grand mean.

Regarding Shukla’s [45] stability variance ($\sigma^{2}$) method:$$\sigma^{2}=\left[\frac{p}{\left(p-2\right)\times (q-1)}\right]W^{2}-\frac{\sum W_{i}^{2}}{\left(p-1\right)\times \left(p-2\right)\times (q-1)}$$where *W^2^* is Wricke’s ecovalence, and p and q are the numbers of genotypes and environments, respectively.

In the Lin and Binns [26] procedure, the index *P_i_* is calculated as follows:$$P_{i}=\sum_{j=1}^{n}{\frac{{\left(X_{ij}-M_{j}\right)}^{2}}{2}}_{n}$$
where *P_i_* is the superiority index of genotype *i*; *X_ij_* is the number of days to anthesis of genotype *i* in environment *j*; *M_j_* is the number of days to anthesis of the genotype with the maximum response among all the genotypes in environment *j*; and *n* is the number of environments.

In the AMMI [22] method:$$Y_{ij}=\mu +\alpha_{i}+\beta_{j}+\sum_{k=1}^{n}\lambda_{k}\gamma_{ik}\delta_{jk}+\rho_{ij}\;\bar{\epsilon_{ij}}$$where $Y_{ij}$ is the mean response of genotype *i* (i = 1, 2, …, g genotypes) in environment *j* (j = 1, 2, …, e environments); μ is the general mean of the experiments; $\alpha_{i}$ is the fixed effect of genotype *i*; *n* is the number of principal axes (principal components) needed to describe the pattern of the interaction between the *i*-eth genotype with the *j*-eth environment; $\beta_{j}$ is the fixed effect of the environment *j*; $\lambda_{k}$ is the *k*-eth single value of the original interactions matrix (called the G X E matrix); $\gamma_{ik}$ is the value corresponding to the *i*-eth genotype in the *k*-eth single vector column of the G × E matrix; $\delta_{jk}$ is the value corresponding to the *j*-eth environment in the *k*-eth single vector (row vector) of the G × E matrix; $\rho_{ij}\;$is the noise associated with the (ga)ij term of the classic interaction of the i-eth genotype with the j-eth environment; and $\bar{\epsilon_{ij}}$ is the mean experimental error.

The WAASB [23] method comprises the following:WAASBG=∑n=1p IPCAgn×EPn∑n−1pEPn
where WAASB*_G_* is the weighted average of the absolute scores of genotype *g*; *IPCAgn* is the score of genotype *g* in the *n*th interaction principal component axis (IPCA); and *EPn* is the amount of the variance explained by the *n*th IPCA.

Individual and joint analyses of variance were performed using SAS PROC GLM [46] at *p* < 0.05. Once the genotype by environment interaction was significant (*p* < 0.05), stability analyses were carried out according to the AMMI (Additive Main Effect and Multiplicative Interaction) [22], Eberhardt and Russell [19], Lin and Binns [26], and WAASB (weighted average of absolute scores from the singular value decomposition of the matrix of the best linear unbiased predictor—BLUPs—for the genotype × environmental—G × E—effects generated by a linear mixed model) [23] protocols. The AMMI and WAASB analyses were run with the R 3.5.2 package metan [47] using the statistical program R Core Team. The Genes statistical software (version 1990. 2023. 92) [48] was used for Eberhardt and Russel’s and Lin and Binns’s analyses.

## 5. Conclusions

Everglades and Simpson had the lowest mean values for days to the first anthesis in all environments being them heat-intolerant. Conversely, BRS Leila, Elisa, and BRS Mediterrânea showed the greatest means to the first anthesis in favorable environments (milder temperatures). Among those cultivars, BRS Mediterânea was the most stable and adapted to unfavorable environments (high temperatures); hence, it could be recommended in high-temperature locations and as a good source for lettuce heat tolerance. The environmental conditions studied in this research, mainly high temperatures, could become a reality in many lettuce-producing areas. Therefore, these results can help to indicate and develop lettuce varieties with greater heat tolerance.

## Figures and Tables

**Figure 1 plants-13-01546-f001:**
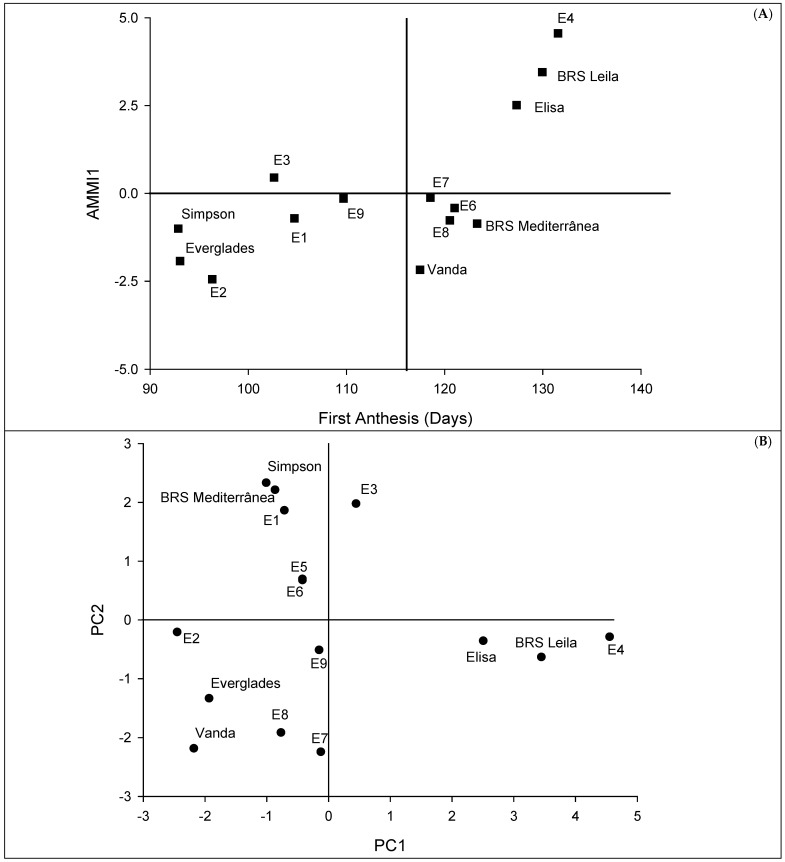
Lettuce heat tolerance, estimated by the number of days to first anthesis, and stability analysis based on (**A**) the AMMI1 biplot and (**B**) the environmental stability GGE biplot. See Table 1 for environment codes.

**Figure 2 plants-13-01546-f002:**
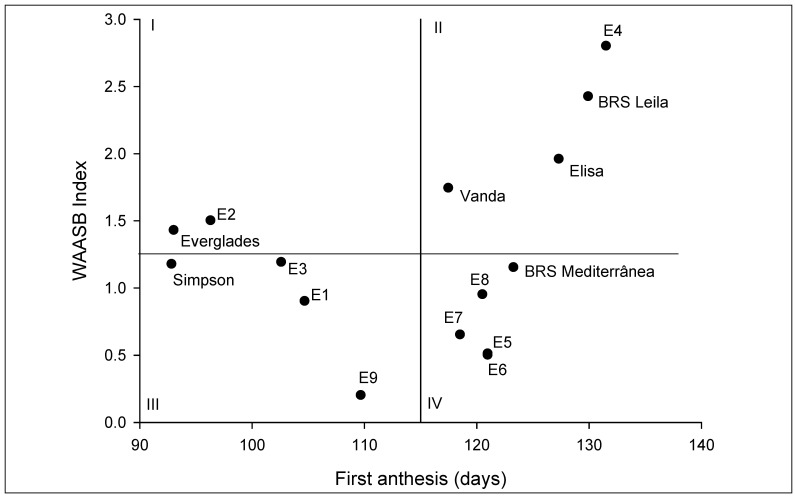
Biplot of lettuce heat tolerance, estimated by the number of days to the first anthesis vs. weighted average of the absolute scores (WAASB Index).

**Table 1 plants-13-01546-t001:** Environments and average temperature of the experimental trials.

Environments (E)	Year	Season	Place	Average Temperature (°C)
E1	2015	Summer	Field	24
E2	2015	Summer	Glass Greenhouse	30
E3	2015	Summer	Plastic Greenhouse	27
E4	2016	Fall	Field	20
E5	2016	Fall	Glass Greenhouse	27
E6	2016	Fall	Plastic Greenhouse	25
E7	2016	Winter	Field	24
E8	2016	Winter	Glass Greenhouse	29
E9	2016	Winter	Plastic Greenhouse	26

**Table 2 plants-13-01546-t002:** AMMI analysis of variance for lettuce heat tolerance estimated by number of days to first anthesis in nine environments during the 2015–2016 agricultural years.

Source of Variation	DF	SS	MS	Cumulative (%)
Replication/Environment	27	2390.04	88.52	
Environment	8	24,369.28	3046.16 **	
Genotype	5	50,989.30	10,197.86 **	
Genotype × Environment	40	5706.40	142.66 **	
PC_1_	12	3232.20	269.35 *	56.60
PC_2_	10	1213.10	121.31 *	77.90
PC_3_	8	999.28	124.91 *	95.40
PC_4_	6	205.80	34.30	99.00
PC_5_	4	56.16	14.04	100.00
Genotype × Environment (linear)	5	1794.10	358.82 **	
Deviation	42	3912.30	93.15 **	
Error	135	1464.75	10.85	

*, ** significant at 5 and 1%, respectively.

**Table 3 plants-13-01546-t003:** Number of days to the first anthesis of six lettuce genotypes across nine environments during the 2015–2016 agricultural years.

Genotypes	Environments ^1^								
	E1	E2	E3	E4	E5	E6	E7	E8	E9
BRS Leila	116 A	106 A	126 A	163 A	134 A	131 A	137 A	137 A	123 A
BRS Mediterrânea	115 A	106 A	120 A	135 B	128 B	133 A	124 B	127 C	117 B
Elisa	117 A	102 A	108 B	157 A	121 C	138 A	131 B	130 B	126 A
Everglades	84 D	81 B	77 C	102 D	97 E	100 C	103 C	101 D	90 C
Simpson Black Seed	92 C	78 B	83 C	106 D	89 F	101 C	91 D	96 E	89 C
Vanda	105 B	105 A	103 B	126 C	117 D	124 B	126 B	132 B	114 B
Average	105	96	103	132	114	121	119	121	110
CV (%)	1.70	2.70	6.02	2.74	1.52	2.27	2.54	1.25	2.91
Environmental Index ^2^	−9.30	−17.67	−11.38	17.54	7.00	7.20	4.54	6.54	−4.29

^1^ Means followed by the same capital letter in columns did not differ in a Skott–Knott test [28] (*p* < 0.05); ^2^ obtained from Eberhardt and Russell.

**Table 4 plants-13-01546-t004:** Stability and adaptability statistics for lettuce heat tolerance estimated by the number of days to the first anthesis in nine environments during the 2015–2016 agricultural years.

Genotypes	First Anthesis (Days)	Eberhardt and Russell			Lin and Binns Modified by Carneiro (Pi’s/10,000)			Wricke’s Ecovalence	Shukla’s Stability Variance	WAASB	Mean Rank
		$b_{i}$	$\sigma_{di}^{2}$	R^2^	P_i_	P_i_+	P_i_−	$W_{i}^{2}$	$\sigma_{i}^{2}$		
BRS Leila	130 (1)	1.30 **	46.51 ** (6)	83.24	4.76 (1)	3.21 (1)	1.54 (1)	285.46 (6)	1739.09 (6)	2.42 (6)	3.5
BRS Mediterrânea	123 (3)	0.80 *	12.18 ** (2)	86.21	7.11 (3)	4.24 (3)	2.87 (3)	76.54 (2)	577.60 (1)	1.15 (1)	2.3
Simpson	93 (6)	0.74 **	8.70 * (1)	87.48	19.92 (6)	12.03 (5)	7.89 (6)	53.36 (1)	591.27 (2)	1.18 (2)	3.6
Elisa	127 (2)	1.45 **	13.67 ** (3)	94.91	5.62 (2)	4.00 (2)	1.63 (2)	79.94 (3)	1284.96 (5)	1.96 (5)	3.0
Everglades	95 (5)	0.84 **	14.93 ** (4)	85.26	19.82 (5)	11.82 (6)	8.00 (5)	83.93 (4)	599.60 (3)	1.43 (3)	4.4
Vanda	118 (4)	0.87 **	27.47 ** (5)	78.43	9.23 (4)	5.37 (4)	3.85 (4)	153.13 (5)	913.96 (4)	1.74 (4)	4.3

*, ** significant at 5 and 1%, respectively.

## Data Availability

The datasets and analysis protocols used during the current study are available from the corresponding author on request. The data are not publicly available due to privacy restriction.

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
