# Peer review of "Stability Evaluation for Heat Tolerance in Lettuce: Implications and Recommendations"

_plants, 2024, doi:10.3390/plants13111546_

Round 1

Reviewer 1 Report

Comments and Suggestions for Authors

The authors report a study on identifying the best cultivars (genotypes) of lettuce resistant to high temperatures.

Although the topic is very important, the manuscript is purely descriptive and hardly innovative. Much work has already been published on the selection of lettuce cultivars resistant to abiotic stress, and in particular, to high temperatures. In order to enrich and make the manuscript more interesting, the authors should have reported molecular, biochemical, and physiological parameters that could explain the mechanisms underlying the high temperature resistance of some lettuce cultivars.

In addition, it would have been interesting to add a genetic characterization that could identify the genes involved in this resistance.

Unfortunately, this manuscript is still in a preliminary stage and needs more data to be accepted by PLANTS.

Comments on the Quality of English Language

Minor editing of English language required

Reviewer 2 Report

Comments and Suggestions for Authors

This manuscript (MS) dissected the genotype by environment interaction (GEI) of six-genotype lettuces for heat tolerance. The results of this MS are solid and valuable for improving lettuce heat tolerance. This MS is well-written and fits the scope of Plants. Some typos should be corrected before publication, such as the 'P<0.05' in Line 21 and the '89 F' in Table 3. Besides, I am curious about the correlation between heat tolerance and flowering time (days to the first anthesis). Do the authors have solid data to indicate their high-reliable relations?

Reviewer 3 Report

Comments and Suggestions for Authors

General comments

This study was to dissect the genotype by environment interaction of lettuce heat tolerance using parametric and nonparametric methods to identify adapted and stable lettuce genotypes.

It is written in acceptable English and generally well organized, including useful findings. However, there are important scientific weaknesses that should be improved.

Below you can find my suggestions.

Abstract: Please organize as follows: Context, Methods, Αρχή φÏŒρμας

Conclusions, Implications. I noticed there are no implications.

L 24. Unsuitable….replace with.. hot and then eliminate ….(high temperatures;

Introduction

L. 93 Usually stability analysis is applied to yield. It is strange to me an application on days to first anthesis. Why you are not using both yield and days to anthesis?

L95. to seek stable and heat-tolerant genotypes of lettuce. Make this research interesting to global interest. For example, the objective was to investigate the relationship between stability, heat tolerance, and productivity. Was the best-performing genotype the most stable in heat tolerance? So, you have yield potential, yield stability, and heat tolerance potential, and heat tolerance stability.

 Materials and Methods

L 265. Please specify the origin of each genotype, these are elite lines hybrids? Genetic origin.

Results

L 100-101. In Table 1 you have summer fall winter please stay on them. Do not put months in the results.

Table 3. Where is LDS? Or any other comparison test?

Discussion

At the end of the day, what are the implications of this research? Should the breeder conduct stability analysis for days before the first flowering?
It is worth it? If we use simple arithmetic means, do we lose important information? Please make this paper important to a global audience.

The title can become more specific. E.g., Stability evaluation for heat tolerance, implications & recommendations.

You concluded that many stability methods should be used. I could not see any reasoning. Where are the major concepts of stability analysis?

Dynamic, static, and a combination of dynamic and static. Heat tolerance is related to a static concept that is also called biological stability. That is the ability of an organism to adapt to the environmental variation.  The cv index, the Shukla's variance, Wricke's ecovalence, and deviations from the joint-regression analysis are good indices.

This is a special case of paper regarding heat tolerance, and you should discuss the static concept of stability.

Round 2

Reviewer 1 Report

Comments and Suggestions for Authors

Dear authors, the critical issues raised in the first review have unfortunately not been resolved.

In fact, the manuscript does not present a defined "scientific question", but only a "descriptive" overview of macroscopic parameters (productivity, drought tolerance). The authors report as the purpose of the paper: "...to investigate the relationship between stability, heat tolerance and productivity", this is descriptive!!! Today, modern scientific research in botanical disciplines needs to answer specific scientific questions. For example: What are the molecular mechanisms underlying the increase or decrease in productivity under abiotic stress conditions in lettuce plants? Which genes or hormones are involved? What is the morphological response of the lettuce leaf? Do stomata increase or decrease? Are there changes in the chlorophyll parenchyma of the lettuce leaf? Or again, do the levels of primary and secondary metabolites change in different lettuce cultivars grown under abiotic stress conditions? I have only given examples of questions; unfortunately, the authors present a manuscript that is still too preliminary. It is not enough to add a few sentences in the introduction or discussion to comment on the critical issues I have raised. In my opinion, further experiments are needed to make this manuscript acceptable for PLANTS. 

Comments on the Quality of English Language

Moderate editing of English language required

Reviewer 3 Report

Comments and Suggestions for Authors

Dear authors, after the first round of review, this manuscript still requires significant improvements to enhance readability. I have annotated numerous comments throughout the text. Based on your data, environments E2 and E8 were the hottest, where "BRS Mediterranea" performed well but was not the best. "Leila" was notably better. It is unclear if stability analysis provided any advantage over mean comparisons. Referring to Table 3, in 7 out of 9 environments, the cultivar "Leila" outperformed Mediterranea. The three nonparametric superiority indices indicate that "Leila" was the most stable across both favorable and unfavorable environments (see Table 4). It is advisable to check whether your data are normally distributed and to identify any outliers. For instance, in E4, there appears to be an anomalously high value. This highlights a limitation of parametric methods, which are sensitive to outliers and require normally distributed data. Therefore, a normality test is essential. Moreover, I am not convinced that "Mediterranea" was superior to "Leida," or possibly "Elisa," as they prevailed in 7 out of 9 environments. The warmer environments are not clearly defined.

Considering Table 3, environments E1, E2, and E3 (summer trials) seem to be unfavorable. No significant differences are evident. "Leida" appears to be broadly adaptable.

For stability analysis, a minimum of 15 environments is recommended, as it was designed for yield trials with more pronounced GxE interactions. In Table 2, the addition of the sum of squares would help determine if Days to Flowering is a genetically controlled trait. If it proves to be heritable, then stability analysis may not be necessary.

Comments on the Quality of English Language

Need editing. 
